# Linking gait mechanics with perceived quality of life and participation after stroke

**David M. Rowland**[1], **Michael D. Lewek**[1,2]*

**1** Department of Health Sciences, Human Movement Science Curriculum, University of North Carolina at Chapel Hill, Chapel Hill, North Carolina, United States of America, **2** Division of Physical Therapy, Department of Health Sciences, University of North Carolina at Chapel Hill, Chapel Hill, North Carolina, United States of America

* mlewek@med.unc.edu

## Abstract

### Background

Individuals with hemiparesis following stroke often experience a decline in the paretic limb's anteriorly directed ground reaction force during walking (i.e., limb propulsive force). Gait speed and walking capacity have been independently associated with paretic limb propulsion, quality of life, and participation in people with stroke. However, it is unclear as to the extent that underlying limb mechanics (i.e., propulsion) play in influencing perceptions of quality of life and participation. We therefore sought to determine the role of limb propulsion during gait on the perception of quality of life and participation in people following stroke.

### Methods

This study is a secondary analysis of individuals involved in a gait retraining randomized control trial. Gait speed, walking capacity, limb propulsion, Stroke Impact Scale, and average daily step counts were assessed prior to and following 6 weeks of training. The pre-training data from 40 individuals were analyzed cross-sectionally using Pearson and Spearman correlations, to evaluate the potential relationship between limb propulsion (ratio of paretic limb propulsion to total propulsion) with gait speed, gait capacity, perceived quality of life domains, and average daily step counts. Partial correlations were used to control for gait speed. Thirty-one individuals were assessed longitudinally for the same relationships.

### Results

We observed a training effect for gait speed, walking capacity, and some quality of life measures. However, after controlling for gait speed, we observed no significant (p≤0.05) correlations in the cross-sectional and longitudinal analyses.

### Significance

After controlling for the influence of gait speed, paretic limb propulsion is not directly related to perceived quality of life or participation. Although limb propulsion may not have a direct effect on participant's perceived quality of life, it appears to be an important factor to

**Data Availability Statement:** All data are now available with de-identified participants within UNC's Dataverse open-source platform: (https://doi.org/10.15139/S3/VEN1RD).

**Funding:** This work was supported by the National Institutes of Health https://www.nih.gov/ (R21 HD068805) to Michael Lewek (MDL) for the original study. The funder did not play any role in the study. The sponsor supported the study decision and preparation of the manuscript.

**Competing interests:** The authors have declared that no competing interests exist.

enhance gait performance, and therefore may be important to target in rehabilitation, when feasible.

## Introduction

Following stroke, individuals often experience persistent unilateral motor deficits (i.e., hemiparesis) [1, 2] into the chronic phase (> six months), resulting in reductions in walking capacity and speed [3, 4] and increases in mechanical work and metabolic cost of walking [5]. Walking capacity and gait speed measures are used to document a person's current and transitory functional recovery status, community mobility and participation level, and quality of life [3, 6, 7]. Given the relationship between gait performance and quality of life [8], it is important to identify the biomechanical mechanisms underlying disordered gait that might contribute to this relationship to ensure that rehabilitation is appropriately targeted.

The presence of motor deficits in the paretic limb following stroke can contribute to deficits throughout the gait cycle (e.g., stance phase stability, limb propulsion, and swing limb advancement). Particularly relevant to the production of forward progression during gait is a reduction in the anteriorly directed ground reaction force (i.e., limb propulsion) [9]. In individuals post-stroke, paretic limb propulsion is positively associated with both gait speed [10] and long-distance walking performance [11]. Additionally, paretic limb propulsion appears to influence a person's community mobility, measured with metrics such as daily step count [12]. Thus, limb propulsion may provide insight into the functional contribution of the affected limb to gait, and may help elucidate whether paretic limb mechanics have recovered versus required compensations [13–15].

Despite deficits in paretic limb propulsion during typical walking, people with stroke exhibit a robust paretic propulsive reserve, suggesting that limb propulsion is modifiable [14]. In fact, the provision of feedback related to gait mechanics can significantly increase limb propulsion [16, 17], suggesting that it may serve as a therapeutic target for patient outcomes. However, a change in limb propulsion with no concomitant change in gait speed/capacity may not be meaningful to people post-stroke. How hard a person pushes off the ground may not lead to a change in mobility if there are no benefits to walking speed/capacity. Paretic limb propulsion is one of many impairments in people with stroke. In this study, we aimed to determine its relative importance beyond the improvements in gait performance.

Given that limb propulsion is related to gait speed and walking capacity [10, 11] and these outcomes are both related to quality of life and participation [3, 4, 6, 7], we sought to investigate the role of limb propulsion as a potential mechanism influencing quality of life measures for people with chronic stroke *directly*. We assessed the potential influence of limb propulsion on quality-of-life measures following stroke both 1) cross-sectionally, and 2) longitudinally. We hypothesized that paretic limb propulsion is associated with quality of life and participation after controlling for walking speed. Likewise, we hypothesized that an improvement in limb propulsion following gait training is associated with quality of life and participation after controlling for changes in walking speed. Knowledge of the relative importance of paretic limb propulsion on quality-of-life and participation measures would inform the relative therapeutic efficacy of targeting limb propulsion in the rehabilitation process. The presence of such a relationship would give merit to targeting gait speed and walking capacity through enhanced paretic limb propulsion rather than targeting other impairments or allowing compensatory strategies to develop.

## Materials and methods

### Participants

Participants in this study were part of a motor learning randomized control trial described elsewhere [18]. Participants were included if they experienced an ischemic or hemorrhagic stroke more than six months prior with resulting asymmetric gait (step length asymmetry index of > 0.537 or stance time asymmetry index of > 0.524) [14]. All included subjects had a comfortable overground gait speed of less than 1.0 m/s, with or without an assistive device and bracing. Participants were excluded if they were concurrently in physical therapy during the study, obtained botulinum toxin to the lower limb within the 6 months leading up to or during the study, had uncontrolled cardiac, respiratory, or metabolic disorders, had neurologic disorders other than stroke, or had a cerebellar lesion. Participants provided informed consent as approved by the Institutional Review Board (IRB) at University of North Carolina at Chapel Hill (IRB # 11–1240). The trial was listed on ClinicalTrials.gov (NCT01598675).

We used the participant's pre-test data to perform a cross-sectional analysis (Cohort 1). Because the hypotheses did not consider the impact of a particular intervention and the primary study did not find significant differences among groups, we chose not to separate participants based on treatment groups. Rather, all participants with data who completed the first training session are included in one group in the cross-sectional analysis (Cohort 1) and all participants that completed the full training are included as one group in the longitudinal analysis (Cohort 2).

### Data collection

We collected outcome measures related to gait speed/walking capacity and quality of life one week prior to training (pre-test), and again one week following the final training session (post-test). We assessed participation from the average daily step count, which is a measure of community engagement and daily activity [3, 19]. We determined the step count from a StepWatch Activity Monitor (Modus Health, Edmonds, WA) that was worn for 4–6 days between the pre-training assessment and the start of training, and from the end of training and the post-training assessment. Comfortable gait speed was measured as participants made three passes across a 14-foot GaitRite pressure mat (CIR Systems, Havertown, PA). Participants were instructed to walk at their preferred, comfortable gait speed, as if they were walking outside of the laboratory setting and not being monitored. We measured walking capacity using the six-minute walk test (6MWT) in which participants were instructed to cover as much distance as possible in six minutes. Participants walked between two tape marks placed 100 feet apart in a hallway as the investigator measured the distance traveled with a measuring wheel. We used the Stroke Impact Scale (SIS) as a quality-of-life measure.

Ground reaction force data from treadmill walking (Bertec Corp, Columbus, OH) were collected at the first training session and during the final (18th) training session. Although these gait training sessions lasted up to 20 minutes, only data from the first two minutes of walking were used. This decision was made because, for two of the three groups, the treadmill belts moved at different speeds from each other starting in minute three. Using data from only the first two minutes ensured that participants always had a belt speed difference (between belts) of zero. Because participants generally improved their walking speed over the 18 sessions of training, all but two subjects in the longitudinal analysis were walking at a faster speed during the final session compared to the initial session.

Gait cycle events (i.e., heel strike and toe off) from the first two minutes of treadmill walking were determined using Visual3D (ver 6, C-Motion, Germantown, MD) and then exported for

further analysis using custom Labview code (National Instruments, Austin, TX). Anterior-posterior ground reaction forces were first standardized by percent bodyweight. Then, ground reaction forces were time normalized to the percent of gait cycle to create an ensemble average [20]. We extracted the propulsive impulse (using 'raw' timeframes) from both the paretic and non-paretic limbs' ensemble averages. Propulsive impulse was calculated as the integral of the anteriorly directed (positive) component of the ensemble averaged time-series. Paretic propulsion impulse ratio was then calculated as a ratio of paretic propulsive impulse divided by the sum of paretic and nonparetic propulsive impulses [9].

## Data analysis

We performed cross-sectional and longitudinal analyses with SPSS (ver 27, IBM, Armonk, NY). Given the known confounding influence of gait speed in limb propulsion, we performed partial correlations, controlling for overground gait speed. For the cross-sectional analysis, we used only the pre-training data from Cohort 1 to perform partial correlations between paretic propulsion impulse ratio and SIS domains most reflective of gait (Activity, Mobility, Participation, Recovery) [21]. We then used pre-training data from Cohort 1 to perform partial correlation between paretic propulsion impulse ratio and average daily step count.

For the longitudinal analysis, we assessed the change in each outcome measure (i.e., limb propulsion, gait speed and 6MWT, SIS domains, average daily step counts) from pre-training to post-training from Cohort 2 with a paired samples t-test. We then used partial correlational analyses (controlled for change in gait speed) to assess for potential relationships between changes in limb propulsion and changes in SIS domains and changes in average daily step counts.

To better visualize the relationship between limb propulsion and participation, we completed a partial regression between paretic impulse ratio and average daily step count in the pre-training data, after the model accounted for gait speed. Additionally, we assessed for a similar relationship with the change variables (i.e., longitudinal analysis) in a similar manner.

## Results

### Participants

Although we had enrolled 48 subjects in the initial study, eight were excluded here from Cohort 1 for incomplete or unusable gait data. Thirty-seven subjects completed training, but six subjects were excluded from Cohort 2 for incomplete or unusable gait data either in the pre-training or post-training data. Two subjects had unusable daily step count data, and were further excluded from Cohort 1 and 2 for those analyses only. The subject's demographics for both Cohort 1 and 2 are found in Table 1.

### Cross sectional analysis

From the pre-training (Cohort 1) data, we observed no significant relationships between paretic impulse ratio and quality of life or participation measures when controlling for gait speed (all $p > .05$, Table 2). This can be further visualized in the partial regression plot between paretic impulse ratio residuals and average daily step count residuals, where there appears to be no significant relationship (Fig 1).

### Longitudinal analysis

Following gait training, the participants in Cohort 2 made improvements in all propulsion measures (all $p \leq .03$, Table 3). Training also yielded a significant improvement in quality-of-

**Table 1. Subject demographics.**

| | Cohort 1 (cross-sectional analysis pre-training, *N* = 40) | Cohort 2 (longitudinal analysis, *N* = 31) |
|---|---|---|
| Sex | 16 F, 24 M | 11 F, 20 M |
| Age, y | 59±11 | 58±11 |
| Height, cm | 170±9 | 170±9 |
| Weight, kg | 81.3±16.5 | 81.7±14.2 |
| Time since stroke, y | 4.7±5.0 | 3.7±3.5 |
| Assistive device | 15 N, 25 Y | 12 N, 19 Y |
| Ankle foot orthosis | 16 N, 24 Y | 15 N, 16 Y |
| Paretic limb | 26 L, 14 R | 18 L, 13 R |
| 6MWT pre-training (m) | 166.1±108.5 | 187.3±108.1 |
| Comfortable gait speed pre-training (m/s) | 0.41±0.23 | 0.45±0.23 |

Abbreviations: *N*, Number; M, Male; F, Female; y, Years; cm, centimeters; kg, kilograms; N, No; Y, Yes; L, Left; R, Right; y, Years; m, meters; m/s, meters per second

life measures, as noted by increases in the SIS Mobility ($p < .01$) and Recovery ($p = .02$) domains. No improvement occurred in SIS Activity ($p = .06$) or Participation ($p = .48$) domains. Additionally, no improvement occurred in average daily step count ($p = 0.51$).

In the longitudinal analysis (Cohort 2), we also observed no significant relationships between change in paretic impulse ratio and change in quality of life or participation measures, when controlling for gait speed (all $p > .05$, Table 4). This is also visualized in the partial regression plot between change in paretic impulse ratio residuals and change in average daily step count residuals, where we do not observe the presence of any significant relationship (Fig 2).

## Discussion

We sought the presence of a potential relationship between limb propulsion with both quality of life and participation measures in people following stroke, to identify a meaningful biomechanical gait target during rehabilitation. Specifically, we had hypothesized that 1) paretic limb propulsion would be associated with participation and perceived quality of life measures, and 2) a change in paretic limb propulsion with gait training would be correlated to a change in participation and quality of life measures. Although paretic limb propulsion is important for walking performance (e.g., gait speed, walking capacity) [19, 22, 23], after we control for walking performance, we failed to observe any significant relationship between limb propulsion and either quality of life or participation. This finding suggests that limb propulsion does not influence perceptions of quality of life and participation beyond that arising from walking performance. This finding was consistent in both a cross-sectional sample, and longitudinally after a 6-week gait training intervention.

Paretic limb propulsion has an established relationship with gait speed [19, 22, 23]. Limb propulsion can accelerate the body forward through manipulations to hip extension power

**Table 2. Partial correlation of propulsion with quality-of-life (Spearman's *Rho*, *N* = 40) and participation (Pearson's *R*, *N* = 38) measures, controlling for comfortable gait speed.**

| | SIS Activity | SIS Mobility | SIS Participation | SIS Recovery | Average Daily Step Count |
|---|---|---|---|---|---|
| Paretic Impulse Ratio (%) | *rho* = .19 | *rho* = -.23 | *rho* = -.02 | *rho* = .01 | *R* = .02 |
| | *p* = .25 | *p* = .16 | *p* = .89 | *p* = .96 | *p* = .89 |

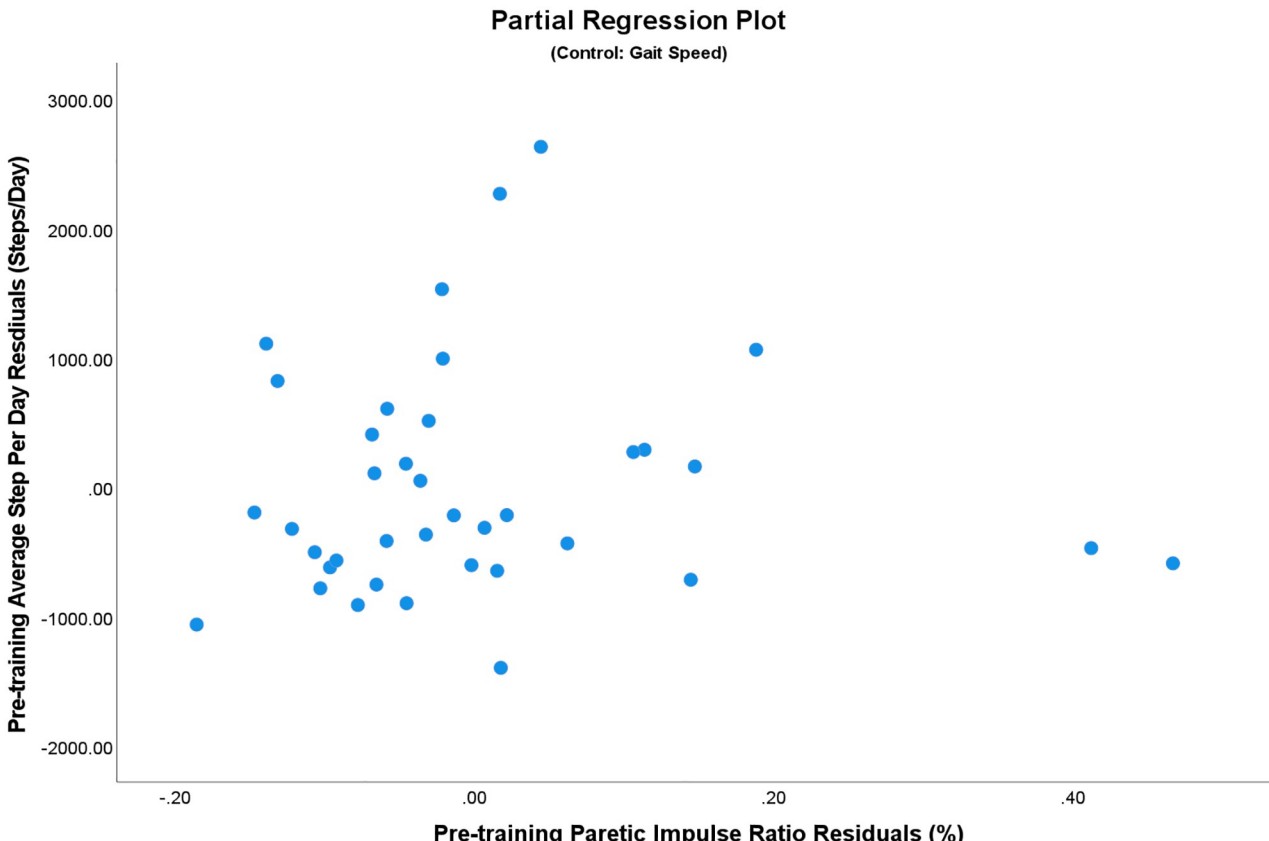

**Fig 1. Partial regression residual plot of pre-training average steps per day against paretic impulse ratio, controlling for comfortable gait speed.**
Despite the two apparent outliers in the lower right quadrant, their removal does not change the results.

[24], ankle plantarflexion power [14, 25], and/or trailing limb posture [14, 25, 26]. Rehabilitation approaches can either target true recovery or how to compensate for reduced limb forces [13]. For example, someone who increases gait speed through enhanced limb propulsion can be considered to have recovered limb function. In particular, people post-stroke may achieve higher gait speeds via enhanced paretic limb propulsion through increased ankle plantarflexor

**Table 3. Paired samples t-tests for outcome measures and propulsion measures (N = 31).**

| Measure | Pre-testing | Post-testing | t | Sig, p (2-tailed) |
|---|---|---|---|---|
| Gait Speed (m/s) | **0.45 ± 0.23** | **0.57 ± 0.27** | **5.82** | **<0 .01** |
| 6MWT (m) | **187.3 ± 108.1** | **223.0 ± 115.8** | **5.54** | **< 0.01** |
| Paretic propulsive impulse (%BW*s) | **1.0 ± 1.1** | **1.7 ± 1.1** | **3.68** | **< 0.01** |
| Nonparetic propulsive impulse (%BW*s) | **4.6 ± 2.8** | **3.3 ± 1.1** | **-2.24** | **0.03** |
| Paretic Impulse Ratio (%) | **19 ± 17** | **32 ± 18** | **4.88** | **< 0.01** |
| SIS Activity | 48.6 ± 14.0 | 51.7 ± 12.6 | 1.94 | 0.06 |
| SIS Mobility | **50.9 ± 14.1** | **60.7 ± 12.3** | **4.33** | **< 0.01** |
| SIS Participation | 38.4 ± 16.6 | 40.8 ± 16.6 | .71 | 0.48 |
| SIS Recovery | **47.7 ± 17.5** | **53.3 ± 16.5** | **2.56** | **0.02** |
| Average Daily Step Count (N = 30) | 1663.0 ± 1230.0 | 1791.5 ± 1161.0 | 1.40 | 0.17 |

Abbreviations: BW, body weight; s, seconds

**Table 4. Partial correlation of change in propulsion with change in quality-of-life (Spearman's *Rho*, N = 31) and participation (Pearson's *R*, N = 30) measures, controlling for change in comfortable gait speed.**

| | ΔSIS Activity | ΔSIS Mobility | ΔSIS Recovery | ΔSIS Participation | ΔAverage Daily Step Count |
|---|---|---|---|---|---|
| ΔParetic Impulse Ratio (%) | *rho* = .27 | *rho* = .22 | *rho* = -.06 | *rho* = .20 | *R* = .16 |
| | *p* = .16 | *p* = .23 | *p* = .78 | *p* = .29 | *p* = .41 |

torque or trailing limb posture [10, 11, 15, 27]. In contrast, a person who uses alternative strategies to increase walking speed has learned to compensate [19]. Compensations may take the form of redistributions of mechanical work to either the contralateral limb [28–30] or to more proximal joints on the paretic limb [31, 32]. Our findings support the idea that people post-stroke perceive their gait performance to be more strongly related to their quality of life, rather than the means by which they achieved that gait performance (i.e., recovery versus compensation of gait speed and walking capacity) [4, 8]. In fact, people with chronic stroke have noted the importance of walking performance, including how far and how fast they can walk, in navigating their daily lives [33].

These data suggest that, while paretic limb propulsion is important for gait and can be a target in rehabilitation, it may not be any more important than other biomechanical components of gait. Instead, the importance to a person's life and participating in activities appears to be driven by walking performance, not the targeted impairment (e.g., propulsion). These data

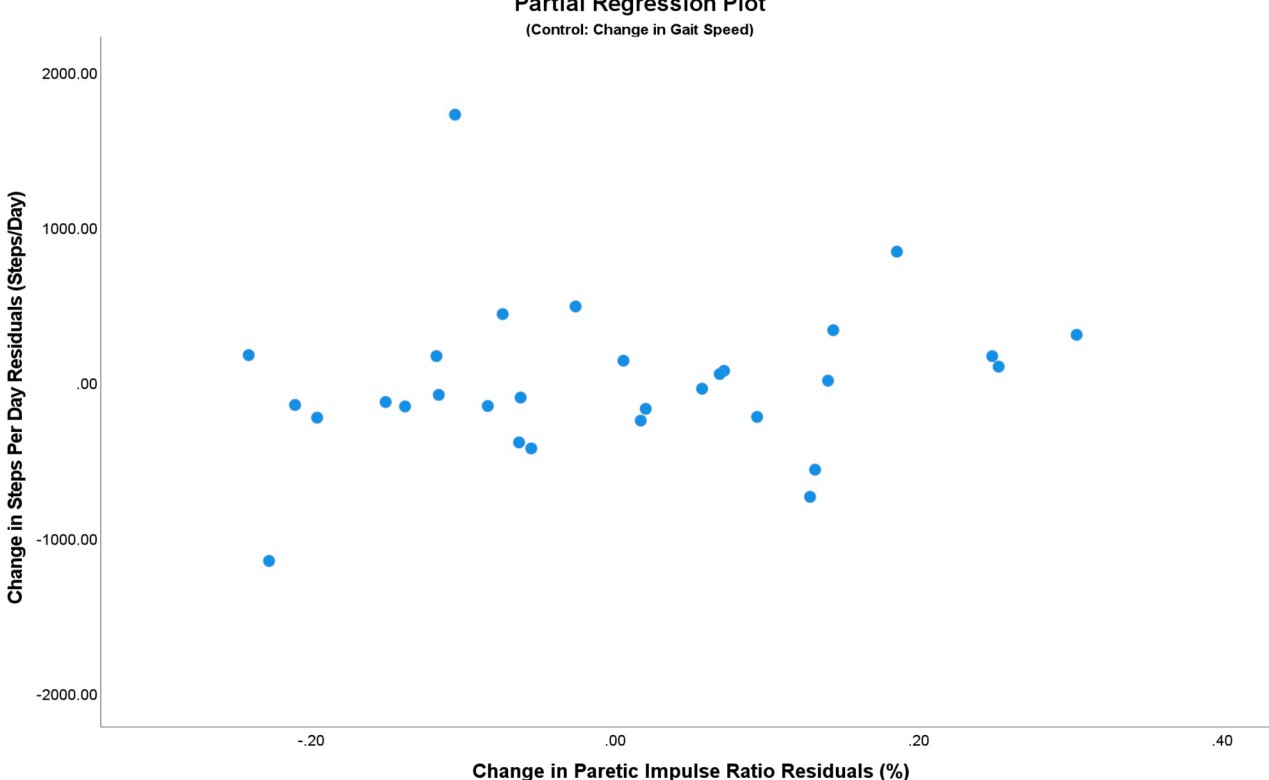

**Fig 2. Partial regression residual plot of change in average steps per day against change in paretic impulse ratio, controlling for change in comfortable gait speed.** Despite the two apparent outliers in the left upper and lower quadrants, their removal does not change the results.

support the framework that while propulsion may be important for improving walking abilities, on its own it is not important for perceptions of quality of life or participation. Rehabilitation practices should be focused on gait performance through task practice [34], which may include propulsion training for improved limb propulsion. However, therapy should not strictly treat observed impairments without considering the overall goal of improved mobility [34].

## Limitations

There are several limitations that need to be acknowledged. First, the gait training intervention was not designed to target limb propulsion, which may have yielded smaller propulsive changes than otherwise expected. Therefore, the changes to limb mechanics may not have been large enough to elicit improvements in quality of life and participation in the longitudinal analysis. Nevertheless, we elicited similar responses in gait speed as others who have explicitly targeted limb propulsion through gait training [35]. Additionally, all but two individuals were walking faster at the end of training in the longitudinal analysis, which may have resulted from improvement in walking mechanics other than paretic limb propulsion. In fact, we observed only small improvements in outcome measures such as gait speed and paretic limb propulsion, suggesting that we may not have elicited large enough changes to gait measures to have an impact on quality-of-life or participation measures. We observed small but significant improvements in gait speed, walking capacity, limb propulsion, and quality of life measures after gait training. Despite improvements within the entire cohort, the observed changes were often small, as only 12 (out of 31) exceeded minimal detectable changes (MDCs) for gait speed [36]. Interestingly, there was no increase in daily step count, suggesting that despite an increase in walking performance, participants did not use this increased function in a meaningful way. Behavioral changes in daily activity may require additional intervention beyond gait training and/or physical therapy. Additionally, we did not observe improvements in several SIS domains (Participation, Activity) after gait training. Although we intentionally chose SIS domains most closely related to gait function and step count as a metric for community and activity participation, we acknowledge that many other factors in individuals' daily lives could influence these measures. Finally, we only assessed for linear relationships, however, it is possible that these data could be subjected to non-linear relationships.

## Conclusions

Paretic limb propulsion does not appear to be important for quality-of-life or participation measures in individuals with chronic hemiparesis following stroke beyond its impact on gait performance. However, this relationship does not reveal that paretic propulsion is not important. Rather, it appears that how hard people with stroke push off the ground does not have an impact on their perceptions or participation in a vacuum. Changes in limb propulsion can influence gait performance, which is important for quality of life and participation. How we achieve this improved gait performance (e.g., through limb propulsion or through other biomechanical targets) does not appear to be important.

## Acknowledgments

We would like to acknowledge Carty Husted, PT, DPT for assistance with participant training; Clint Wutzke, PhD for assistance with data collection, and Rob Sykes III, PT, DPT for assistance with data management.

## Author Contributions

**Conceptualization:** David M. Rowland, Michael D. Lewek.

**Data curation:** David M. Rowland.

**Formal analysis:** David M. Rowland, Michael D. Lewek.

**Investigation:** Michael D. Lewek.

**Methodology:** David M. Rowland, Michael D. Lewek.

**Project administration:** Michael D. Lewek.

**Software:** David M. Rowland, Michael D. Lewek.

**Supervision:** Michael D. Lewek.

**Validation:** Michael D. Lewek.

**Visualization:** David M. Rowland.

**Writing – original draft:** David M. Rowland.

**Writing – review & editing:** David M. Rowland, Michael D. Lewek.

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
