## [Decision Letter · Decision Letter 0]

29 Jun 2022

PONE-D-22-07362Linking gait mechanics with perceived quality of life and participation after strokePLOS ONE

Dear Dr. Rowland,

Thank you for submitting your manuscript to PLOS ONE. After careful consideration, we feel that it has merit but does not fully meet PLOS ONE’s publication criteria as it currently stands. Therefore, we invite you to submit a revised version of the manuscript that addresses the points raised during the review process.

 As you can see below, the reviewers were positive on your work. Please, reply carefully all questions raised by the reviewers.

We look forward to receiving your revised manuscript.

Kind regards,

Leonardo A. Peyré-Tartaruga, Ph.D.

Academic Editor

PLOS ONE

Journal Requirements:

2. We noted in your submission details that a portion of your manuscript may have been presented or published elsewhere. Please clarify whether this publication was peer-reviewed and formally published. If this work was previously peer-reviewed and published, in the cover letter please provide the reason that this work does not constitute dual publication and should be included in the current manuscript.

Reviewers' comments:

Reviewer's Responses to Questions

**Comments to the Author**

1. Is the manuscript technically sound, and do the data support the conclusions?

Reviewer #1: Yes

Reviewer #2: Yes

Reviewer #3: Yes

2. Has the statistical analysis been performed appropriately and rigorously? 

Reviewer #1: Yes

Reviewer #2: I Don't Know

Reviewer #3: Yes

3. Have the authors made all data underlying the findings in their manuscript fully available?

Reviewer #1: Yes

Reviewer #2: No

Reviewer #3: Yes

4. Is the manuscript presented in an intelligible fashion and written in standard English?

Reviewer #1: Yes

Reviewer #2: Yes

Reviewer #3: Yes

5. Review Comments to the Author

Reviewer #1: The article is interesting and well-written. Moreover, the statistical analysis il well-conducted. I have only the following comments.

1. The statistical methods considered by the authors rely on assumptions about the nature of the underlying data. If the data do not meet those assumptions, then the results often are not valid. Therefore, it is important for authors to check that those assumptions are satisfied for the data at hand, at least, approximately.

2. Figures 1 and 2 need to be further commented. Just as an example, the shape of the scatter plot in Figure 1 is “particular” (it seems there is more than one group and leverage points) while from Figure 2 it seems to me that there are some outliers.

3. All the statistical symbols should be: 1) defined for the reader the first time they are used and 2) written in italic. Just as an example, define N at line 97.

4. Consider including the following paper in your introduction and discussion (PMCID: 
 PMC7390624 ).

Reviewer #2: The authors present a study that indicates that it is part or segment of another study, and the same number is observed in the clinical trial registration. Which in no way diminishes the scientific relevance. The complementary data informing how the training protocols were performed are clear in the studies mentioned as the basis for this one. It is suggested, however, that the subjects' database (without naming them) and collection data be made available in complementary material.

Reviewer #3: The present study determines the role of limb propulsion during gait on the perception of quality of life and participation in people following stroke. After controlling for gait speed, paretic limp propulsion is not correlated to perceived quality of life or participation. Although may not have a related effect, limp propulsion appears to be an important factor to enhance gait performance, and this highlights the importance of further investigations in the rehabilitation area.

Below, some considerations:

An important factor was presented in the introduction and deserves discussion: a change in limb propulsion without a concomitant change in speed or gait ability may not be representative for people after stroke. How hard a person pushes the ground may not lead to a change in mobility if it is not related to benefits in speed or walking ability. In this scenario, the self-selected walking speed and maximum walking speed bring us a better understanding of the functional improvement of gait, as it is related to both energy and mechanical efficiency. (https://doi.org/10.1016/j.jstrokecerebrovasdis.2021.106023 - “Comfortable and Maximum Gait Speed in Individuals with Chronic Stroke and Community-Dwelling Controls” and 10.4103/2468-5658.184750 – “A new integrative approach to evaluate pathological gait: locomotor rehabilitation index”)

#Materials and Methods

*Participants

Page 5, Line 97-105: This section is specifically to describe materials and methods used in the research, as well as to describe participants characteristics, without any results on the number of included/excluded. You can describe that you used pre-test data to perform the cross-sectional analysis, however, I suggest not putting the N as well for the post-training analysis. I suggest leaving all results in relation to N included/excluded to the RESULTS section, presenting a flowchart for both cross-sectional and post-training analysis, it would be clearer for readers. About table 1, it should also be placed in the RESULTS section.

#Data Collection

Page 6, Line 118-119: Please, this information refers to the result, insert in the RESULTS section.

Page 7, Line 131-133: Please enter this information in the RESULTS section.

Page 8, Line 148-149: Please delete the N=40 information, just leave it described Cohort 1 so readers will know what it is.

Page 8, Line 155: “(N = 29 for step count, N = 31 for all others)”, you can remove this information, leave it to the RESULTS section.

Page 8, Line 161: “(N=38), remove this information, leave it to the RESULTS section.

Page 8, Line 162-163: “(N=29), remove this information, leave it to the RESULTS section.

*Is comfortable gait speed equivalent to self-selected walking speed? I ask this question because it is described like this: “Comfortable gait speed was measured as participants made three passes across a 14-foot GaitRite pressure mat (CIR Systems, Havertown, PA)”. Were participants instructed to walk at the most comfortable speed? This is a very important factor as they can often walk slower or faster than their usual comfortable speed during a test.

RESULTS

Participants reduced an average of 125 steps for the Step Count variable after the training period. I believe that 125 steps in a stroke population is relevant. Can you discuss this and bring possible explanations for this fact? Since they have improved walking speed and walking ability, why have they reduced the number of steps? I would expect the number of steps to increase with rehab. For this aspect, it would be interesting to calculate the effect size and see the magnitude of this result, because apparently 125 steps less are clinically relevant in my opinion, despite not having presented a statistically significant difference, the p value is very close to that.

DISCUSSION

Page 12, Line 213-221: In this scenario, your training was performed only on the treadmill. I believe it will be interesting for the future to investigate precisely the effect of specific strength exercises to improve limb propulsion, as well as trunk and balance postural control exercises. I fully agree with the rationale that “For example, someone who increases gait speed through enhanced limb propulsion can be considered to have recovered limb function”, however, it is possible to improve gait speed without improving limb function. Would it be more important to worry about gait speed, as it would be more related to functionality? This discussion is very important, because according to the findings of the present study, they perceive that gait performance is strongly related to quality of life, rather than the means by which they achieved, and in my opinion, the quality of life of individuals is very important.

6. PLOS authors have the option to publish the peer review history of their article (what does this mean?). If published, this will include your full peer review and any attached files.

Reviewer #1: No

Reviewer #2: No

Reviewer #3: **Yes: **Leandro Tolfo Franzoni

---

## [Author Response · Author response to Decision Letter 0]

28 Jul 2022

Comments are attached below. For a more clear response, we have attached a Response to Reviewers as a .doc file, with red text and indenting to highlight our responses. It is included below as well. 

We thank the reviewers for their thoughtful critique of our manuscript. We have responded to each of the comments below on a point-by-point basis. Each of the reviewers comments is in italics, with our response directly below.

EDITOR’S COMMENTS:

We have updated the reference list to reflect the correct style (Vancouver), and the title page to reflect the correct formatting. We checked and verified the main body to ensure that it corresponds to the correct headings and sizes. We moved the ethics statement to the Materials and methods section. The figures are appropriately labeled as additional file formats. 

2. We noted in your submission details that a portion of your manuscript may have been presented or published elsewhere. Please clarify whether this publication was peer-reviewed and formally published. If this work was previously peer-reviewed and published, in the cover letter please provide the reason that this work does not constitute dual publication and should be included in the current manuscript.

These data have not been published previously. The work here represents an analysis of previously unpublished secondary outcome measures from a clinical trial. The conclusions were presented at the American Physical Therapy Association’s Combined Sections Meeting (CSM) as a 10-minute platform talk. This was accompanied by an abstract, but a full paper has not been published with these findings. Therefore there is no risk of dual publication.

All data are now available with de-identified participants within UNC’s Dataverse open-source platform: (https://doi.org/10.15139/S3/VEN1RD)

Modified as requested. 

Reference list has been checked, and updated to reflect the proper format (Vancouver). 

REVIEWER #1

Reviewer #1: The article is interesting and well-written. Moreover, the statistical analysis is well-conducted. I have only the following comments.

1. The statistical methods considered by the authors rely on assumptions about the nature of the underlying data. If the data do not meet those assumptions, then the results often are not valid. Therefore, it is important for authors to check that those assumptions are satisfied for the data at hand, at least, approximately.

As requested, we have verified that our dataset met the six standard assumptions for partial regression. 

2. Figures 1 and 2 need to be further commented. Just as an example, the shape of the scatter plot in Figure 1 is “particular” (it seems there is more than one group and leverage points) while from Figure 2 it seems to me that there are some outliers.

We agree that Figure 1 appears to have two ‘leverage’ points. To test this, we removed the two points on the lower right quadrant (i.e., those > 0.40 % Impulse Ratio residuals), which revealed no change in the statistical outcome or interpretation of the data. Likewise, we used Cook’s distance to determine that those two data points were not considered outliers (Cook’s distance = 0.20 and 0.33, respectively for the two data points). Similarly, in Fig 2, we removed apparently outliers/leverage points and, again, no change in the interpretation of the data or significance was noted. As a result, we have opted to leave these data points in the analysis with a note in the Figure captions to indicate to the reader that we considered the possibility that the data points were outliers and/or leverage points, but our analysis rejected that contention. 

3. All the statistical symbols should be: 1) defined for the reader the first time they are used and 2) written in italic. Just as an example, define N at line 97.

Modified as suggested. 

4. Consider including the following paper in your introduction and discussion (PMCID: PMC7390624 ).

Included as suggested. 

REVIEWER #2

Reviewer #2: The authors present a study that indicates that it is part or segment of another study, and the same number is observed in the clinical trial registration. Which in no way diminishes the scientific relevance. The complementary data informing how the training protocols were performed are clear in the studies mentioned as the basis for this one. It is suggested, however, that the subjects' database (without naming them) and collection data be made available in complementary material.

Please see response above regarding lack of dual-publication, and how we have now made data available.

REVIEWER #3

Reviewer #3: The present study determines the role of limb propulsion during gait on the perception of quality of life and participation in people following stroke. After controlling for gait speed, paretic limp propulsion is not correlated to perceived quality of life or participation. Although may not have a related effect, limp propulsion appears to be an important factor to enhance gait performance, and this highlights the importance of further investigations in the rehabilitation area.

Below, some considerations:

An important factor was presented in the introduction and deserves discussion: a change in limb propulsion without a concomitant change in speed or gait ability may not be representative for people after stroke. How hard a person pushes the ground may not lead to a change in mobility if it is not related to benefits in speed or walking ability. In this scenario, the self-selected walking speed and maximum walking speed bring us a better understanding of the functional improvement of gait, as it is related to both energy and mechanical efficiency. (https://doi.org/10.1016/j.jstrokecerebrovasdis.2021.106023 - “Comfortable and Maximum Gait Speed in Individuals with Chronic Stroke and Community-Dwelling Controls” and 10.4103/2468-5658.184750 – “A new integrative approach to evaluate pathological gait: locomotor rehabilitation index”)

We agree with the reviewer, as this was the aim of our paper. Our conclusions coincide with the reviewer’s comment regarding the fact that how a person achieves increased gait speed really should not matter, but that they are able to do so. This point is discussed in the Discussion section, to tie together the point that propulsion only matters if it leads to a benefit to gait speed/walking capacity. 

#Materials and Methods

*Participants

Page 5, Line 97-105: This section is specifically to describe materials and methods used in the research, as well as to describe participants characteristics, without any results on the number of included/excluded. You can describe that you used pre-test data to perform the cross-sectional analysis, however, I suggest not putting the N as well for the post-training analysis. I suggest leaving all results in relation to N included/excluded to the RESULTS section, presenting a flowchart for both cross-sectional and post-training analysis, it would be clearer for readers. About table 1, it should also be placed in the RESULTS section.

As recommended, we have now left only the description of the participant characteristics, and have moved the numbers (sizes) of the cohorts to the results section. 

#Data Collection

Page 6, Line 118-119: Please, this information refers to the result, insert in the RESULTS section.

Page 7, Line 131-133: Please enter this information in the RESULTS section.

Page 8, Line 148-149: Please delete the N=40 information, just leave it described Cohort 1 so readers will know what it is.

Page 8, Line 155: “(N = 29 for step count, N = 31 for all others)”, you can remove this information, leave it to the RESULTS section.

Page 8, Line 161: “(N=38), remove this information, leave it to the RESULTS section.

Page 8, Line 162-163: “(N=29), remove this information, leave it to the RESULTS section.

All above have been integrated into the paper and corrected. 

*Is comfortable gait speed equivalent to self-selected walking speed? I ask this question because it is described like this: “Comfortable gait speed was measured as participants made three passes across a 14-foot GaitRite pressure mat (CIR Systems, Havertown, PA)”. Were participants instructed to walk at the most comfortable speed? This is a very important factor as they can often walk slower or faster than their usual comfortable speed during a test.

This has been clarified in this section – participants were instructed to walk at a self-selected comfortable pace. Although not reported, after recording the comfortable pace, we asked participants to self-select their fastest (but safe) gait speed. Thus, we are well aware that self-selected speeds can range from slow to fast. Here, however, we only report on participant’s self-selected comfortable gait speed. 

RESULTS

Participants reduced an average of 125 steps for the Step Count variable after the training period. I believe that 125 steps in a stroke population is relevant. Can you discuss this and bring possible explanations for this fact? Since they have improved walking speed and walking ability, why have they reduced the number of steps? I would expect the number of steps to increase with rehab. 

For this aspect, it would be interesting to calculate the effect size and see the magnitude of this result, because apparently 125 steps less are clinically relevant in my opinion, despite not having presented a statistically significant difference, the p value is very close to that.

We are grateful that the reviewer brought up this point as it caused us to go back to our original dataset to determine why step count decreased. We had inadvertently pulled data cells from the wrong column initially. The data have been checked and double-checked at this point and we are confident that the data are accurate. With the correct numbers, we see a non-significant increase in daily step counts of ~128 steps/day. This change is fairly small with a small effect size (0.256). We regret the mistake, but are grateful that the reviewer alerted us to the possibility of this error. All analyses have been rerun, and tables/figures/text updated with the correct values.

DISCUSSION

Page 12, Line 213-221: In this scenario, your training was performed only on the treadmill. I believe it will be interesting for the future to investigate precisely the effect of specific strength exercises to improve limb propulsion, as well as trunk and balance postural control exercises. I fully agree with the rationale that “For example, someone who increases gait speed through enhanced limb propulsion can be considered to have recovered limb function”, however, it is possible to improve gait speed without improving limb function. Would it be more important to worry about gait speed, as it would be more related to functionality? This discussion is very important, because according to the findings of the present study, they perceive that gait performance is strongly related to quality of life, rather than the means by which they achieved, and in my opinion, the quality of life of individuals is very important.

We are happy that the reviewer agrees with our statements and note that the reviewer’s points are well illustrated in our Discussion section: “Our findings support the idea that people post-stroke perceive their gait performance to be more strongly related to their quality of life, rather than the means by which they achieved that gait performance (i.e., recovery versus compensation of gait speed and walking capacity).” and “while paretic limb propulsion is important factor for gait and can be a target in rehabilitation, it may not be any more important than other biomechanical subcomponents of gait.”

---

## [Editor Report · Decision Letter 1]

30 Aug 2022

Linking gait mechanics with perceived quality of life and participation after stroke

PONE-D-22-07362R1

Dear Dr. Rowland,

We’re pleased to inform you that your manuscript has been judged scientifically suitable for publication and will be formally accepted for publication once it meets all outstanding technical requirements.

Kind regards,

Leonardo A. Peyré-Tartaruga, Ph.D.

Academic Editor

PLOS ONE

---

## [Editor Report · Acceptance letter]

1 Sep 2022

PONE-D-22-07362R1 

Linking gait mechanics with perceived quality of life and participation after stroke 

Dear Dr. Rowland:

I'm pleased to inform you that your manuscript has been deemed suitable for publication in PLOS ONE. Congratulations! Your manuscript is now with our production department. 

Kind regards, 

on behalf of

Professor Leonardo A. Peyré-Tartaruga 

Academic Editor

PLOS ONE